# THE AGENT'S MARATHON: PROBING THE LIMITS OF ENDURANCE IN LONG-HORIZON TASKS

## ABSTRACT

Large Language Model (LLM) agents, augmented with diverse tools, have shown impressive progress in domains such as scientific discovery and enterprise automation. Yet they remain brittle in long-horizon tasks that require extended sequences of interactions, where performance often deteriorates rapidly. Existing benchmarks provide only partial coverage of this challenge: manual or crowdsourced tasks are too short, tool-use benchmarks emphasize breadth over depth, and web-based evaluations rely on emergent rather than controllable complexity. To fill this gap, we introduce TaskWeaver, a rule-based, controllable platform for generating benchmark tasks with precisely adjustable difficulty and horizon length. At its core, TaskWeaver abstracts all tool use as file-read operations. This design choice removes superficial API complexities, allowing us to directly probe an agent's core ability to reason and integrate intermediate results over long, dependent sequences. We instantiate the framework across three domains: document understanding and navigation, multi-modal information integration, and executable code analysis. Each domain probes a complementary aspect of agentic reasoning, and together they form a unified benchmark, LORE (Long-horizon Reasoning Evaluation). Empirical results show that even for the strongest models we tested, performance degrades significantly as task length and per-step complexity increase. Specifically, their accuracy approaches zero on tasks exceeding 120 steps, and on more challenging variants, performance collapses in fewer than 15 steps. These findings highlight long-horizon robustness as a central open challenge for future agent development.

## 1 INTRODUCTION

Modern Large Language Model (LLM) agents, augmented with diverse tools, offer rich possibilities for interacting with real-world data and systems, and they have already demonstrated success across applications such as scientific discovery and enterprise automation (Schmidgall et al., 2025; Luo et al., 2025). Despite this, agents face a critical challenge of performance degradation in long-horizon tasks that require a high number of sequential interactions (Wang et al., 2025b; Erdogan et al., 2025). In such settings, failure often comes not only from poor planning but also from small execution mistakes, which can add up and cause the whole task to fail (Sinha et al., 2025; Zhou et al., 2025).

However, existing benchmarks do not adequately evaluate an agent's robustness to error accumulation over long-horizon tasks. Specifically, mainstream agentic benchmarks fall into three categories. The first class, represented by GAIA (Mialon et al., 2023) and HLE (Phan et al., 2025), relies on tasks that are manually designed or crowdsourced. The high cost of this annotation process inherently limits the length and logical depth of the tasks. For instance, in GAIA, the majority of human-annotated solution paths are concentrated within 10-20 steps. The second class, including AgentBench (Liu et al., 2024), API-Bank (Li et al., 2023) and ToolBench (Qin et al., 2024), focuses more on the breadth of tool use. Their design prioritizes evaluating an agent's ability to handle diverse APIs or a wide range of executable tools, rather than its performance on deep, sequential reasoning within a single task. The third category, exemplified by WebArena (Zhou et al., 2024) and Mind2Web (Deng et al., 2023), tests an agent's ability to decompose problems and execute actions within realistic web environments. Although these benchmarks may contain tasks described as "long-horizon," their length and complexity are often emergent properties of intricate web UIs or complex API combinations, rather than a controllable, arbitrarily extendable chain of logical dependencies.

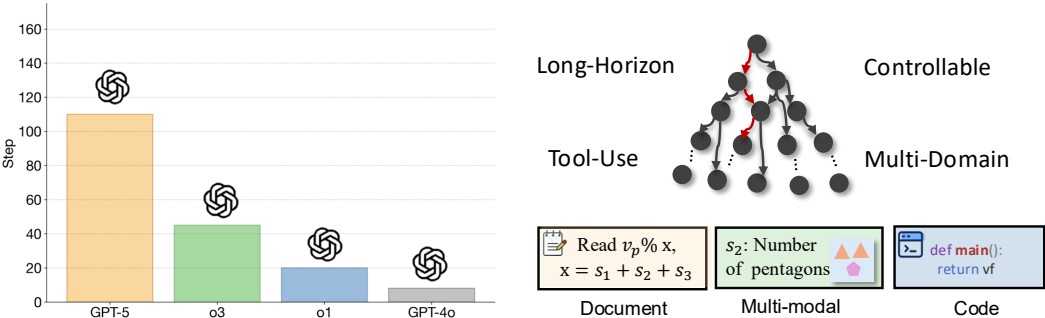

Figure 1: The challenge of long-horizon tasks for current AI agents and our benchmark LORE designed to evaluate it. *Left*: Even state-of-the-art language models are brittle on tasks requiring long sequences of reasoning. Their performance, measured by the number of successful steps (*y-axis*), degrades sharply as the task horizon extends, highlighting the critical need to improve their long-horizon robustness. *Right*: An overview of our proposed benchmark framework, which is designed to be long-horizon and controllable. It integrates multi-domain tasks, including document understanding, multi-modal reasoning, and code analysis, under a unified tool-use paradigm to precisely evaluate and advance the core reasoning and planning capabilities of AI agents.

To address this gap, as illustrated in Figure 1, we introduce **TaskWeaver**, a rule-based, controllable platform for generating benchmark tasks that probe the long-horizon reasoning of LLM agents. TaskWeaver can produce an unlimited number of verifiable evaluation tasks whose difficulty and horizon length are precisely controlled, and it naturally scales to new task schemas. At its core lies a unifying abstraction: each tool call is treated as a file-read operation. This abstraction is natural for two reasons. First, from the perspective of the agent, a tool invocation is essentially a black-box I/O: given a query, the model receives a textual result that is appended to its context, much like retrieving the contents of a file. Second, this view removes the superficial characteristics of diverse APIs while preserving the essential property we aim to evaluate, namely the agent's ability to track state, integrate intermediate results, and reason across long sequences of dependent steps. By reducing tool use to file reads, we retain the fundamental challenge of long-horizon reasoning without confounding factors such as memorizing API syntax or handling heterogeneous interfaces.

Building upon this, we curate our benchmark called **LORE** (**LO**ng-horizon **R**easoning **E**valuation) across three domains: (1) a foundational text-based domain that tests an agent's ability to synthesize information from a multi-document system; (2) a multi-modal extension that requires the integrated analysis of textual and visual data; and (3) a complex code analysis domain designed to stress-test an agent's advanced reasoning capabilities on intricate program logic and control flow. Our evaluation across these domains reveals several limitations in current agent capabilities. First, the performance of state-of-the-art LLMs degrades with task length, with accuracy approaching zero on tasks exceeding around 120 steps. Second, this decay accelerates as per-step difficulty increases; on our more challenging benchmark variants, agent performance collapses after fewer than 15 steps. These evaluation results indicate critical limitations in the long-horizon capabilities of current state-of-the-art LLM agents.

## 2 RELATED WORK

### 2.1 LLM AGENT BENCHMARKS

The evaluation of LLM agents has spurred a variety of benchmarks, each with specific focuses and limitations. For instance, GAIA (Mialon et al., 2023) and HLE (Phan et al., 2025) uses real-world questions, but its tasks are too short to effectively measure long-horizon planning (Erdogan et al., 2025). Benchmarks like AgentBench (Liu et al., 2024), WebArena (Zhou et al., 2024) and Mind2Web (Deng et al., 2023) immerse agents in complex, multi-turn environments, but their emergent dependency structures prevent the controlled measurement of reasoning endurance. Another line of work focuses on interaction dynamics. MINT (Wang et al., 2023) assesses the use of tools and feedback, and AgentBoard (Chang et al., 2024) introduces new analytical metrics. These prioritize

the interactive process or evaluation methodology over designing tasks that specifically stress-test an agent's autonomous reasoning endurance. While ToolBench (Qin et al., 2024) and its successor StableToolBench (Guo et al., 2024) and API-Bank (Li et al., 2023) test the use of many APIs, their tasks require only simple tool compositions (Peng et al., 2022; Huang et al., 2024; Wang et al., 2025a), falling short of the deeply nested dependency chains designed in LORE (Yao et al., 2024).

## 2.2 Synthetic and Procedural Benchmark Generation

Our work is built upon the idea of automatic data generation, drawing from the established field of Procedural Content Generation (PCG) and contrasting with recent efforts in automated agent task generation (Maheshwari et al., 2024; Peper et al., 2025; Long et al., 2024; Wen et al., 2024). Procedural Content Generation (PCG) is a mature field, primarily from game development, that uses algorithms to create content rather than manual creation (Khalifa et al., 2025). The core principles of PCG, controllability, quality, and diversity, directly map to the design goals of TaskWeaver. Recent work has also explored automating agent benchmark generation (Patel et al., 2025; Butt et al., 2024; Nathani et al., 2025; Zhuge et al., 2024). TaskCraft (Shi et al., 2025) proposes a workflow that starts with simple "atomic" single-tool tasks and increases complexity through "depth-based" (serial) and "width-based" (parallel) extensions (Shi et al., 2025). OdysseyBench (Wang et al., 2025b) uses a multi-agent framework to synthesize long-horizon workflows by simulating multi-day user-assistant dialogues in office application environments (Wang et al., 2025b). TaskWeaver differs fundamentally by employing a *first-principles generative* approach. It starts from a final answer (the root of a tree) and works backward, applying formal operations to generate the entire dependency tree. This method is uniquely suited for testing abstract, domain-agnostic reasoning by directly generating the formal reasoning structures themselves.

## 3 TASKWEAVER FRAMEWORK

TaskWeaver introduces a procedural generation framework for constructing complex, long-horizon tasks with controllable difficulty. At its core is a unifying abstraction that treats each tool use as a single, analyzable file-read operation. Building on this abstraction, we construct a dependency graph in tree form and iteratively expand it with a predefined set of formal operators, enabling the creation of tasks of arbitrary length and logical complexity. In the following subsections, we first define the key components of our framework, including the tool-call abstraction and operator set. We then detail the iterative task generation process, which supports both bottom-up and top-down construction. Finally, we show how this general framework instantiate benchmarks in three domains: document understanding and navigation, multimodal information analysis, and code understanding.

### 3.1 Preliminary Concepts

Our TaskWeaver framework is built upon three core concepts: the abstraction of tool calls, the use of a tree structure to represent dependencies, and a formal definition for generative operators.

**Unified Tool Interaction Abstraction.** We unify diverse tool interactions under a common abstraction: each tool call is treated as a file-read operation. Concretely, an agent provides information to a tool and receives a textual output, much like reading the contents of a file given its identifier. This view strips away the surface differences among APIs and instead emphasizes the essential property we aim to evaluate, namely the agent's ability to track and reason over long chains of stateful dependencies. Throughout the framework, we therefore treat every tool call as a standardized file-read operation, which serves as the basic unit for constructing the dependency tree.

**Dependency Tree Representation.** To represent the dependencies among tool calls, we organize them into a tree $\mathcal{T} = (V, E)$, where $V$ and $E$ are the set of nodes and edges, respectively. Each node corresponds to a single call, modeled as a "file" produced by the tool. A node $v \in V$ is denoted as a tuple $v = (id, c)$, where $id$ is a unique tool identifier, $c$ is the raw content of the tool's output. The edges $E$ capture dataflow dependencies: the content of a parent node can be derived from the information in its children. This representation explicitly records how intermediate results combine to form higher-level outputs, making long reasoning chains both verifiable and analyzable. For example, given the query '*What will the temperature be in New York tomorrow?*', a call to `WeatherAPI("New York, tomorrow")` can be represented as node $v_1$ with $id =$

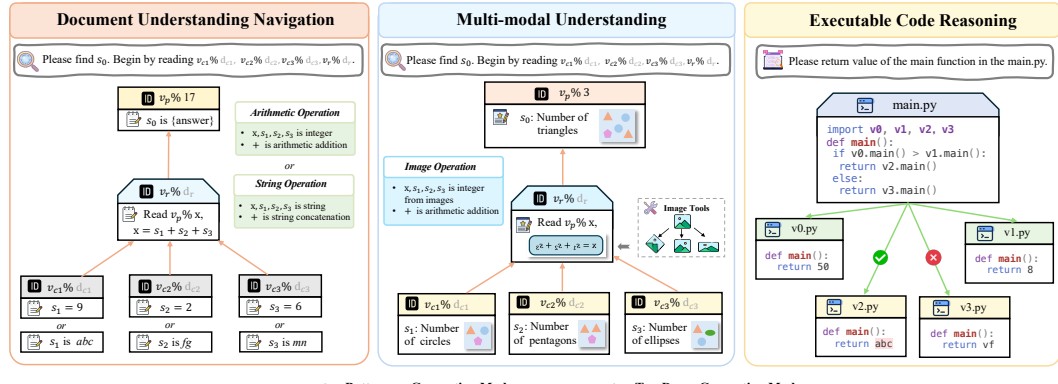

Figure 2: Concrete examples of tasks generated by TaskWeaver, showcasing the construction of dependency trees in three domains. *Document Understanding Navigation* demonstrates tasks requiring retrieval and calculation with either numbers or strings. *Multi-modal Understanding* shows a task where reasoning requires extracting information from images. *Executable Code Reasoning* presents a task where the agent must trace function calls and conditional logic across multiple files. The structure of each task defines a verifiable reasoning path, which can be generated using either a bottom-up approach (orange arrows) where content flows from leaves to the root, or a top-down approach (green arrows) where the task is decomposed from the root to the leaves.

WeatherAPI and $c$ = the output. A second node $v_2$, with $id$ = 'ExtractTemperature' and $c$ = the numeric value, depends on $v_1$. Here $id$ indicates the tool invoked, $c$ the returned content, and the edge captures their dependency.

**Task Expansion Rules.** The dependency tree grows through a pre-defined set of generative operators $\mathcal{O}$. Each operator $o \in \mathcal{O}$ is a formal rule that specifies how to expand a single node into a new parent with one or more children. An operator defines its arity (the number of children it can generate, within a specified range), may enforce type constraints on the parent and child nodes, and provides procedures for creating new children, generating the rule that links them, and executing that rule to produce the parent's content. This mechanism ensures that each expansion step is well-defined and consistent, allowing tasks of arbitrary length and complexity to be generated in a systematic way.

For example, extending the weather query from above, the node $v_2$ that extracts the temperature value can be further expanded by applying an operator ConvertUnits. This operator takes $v_2$ as input and generates a new parent node $v_3$ with $id$ = "ConvertUnits" and $c$ = the temperature converted from Fahrenheit to Celsius. Here, the operator formally specifies both the child instantiation (create a numeric child node) and the rule execution (apply the conversion formula). The resulting tree makes explicit that the Celsius value depends on the extracted Fahrenheit value, which in turn depends on the raw API response.

## 3.2 TASK GENERATION PROCESS

Building on the preliminary concepts, we now describe the process for generating task instances through the construction of a dependency tree. The generation begins with an empty root node and proceeds in an iterative manner: at each step, a leaf node is selected and expanded using a randomly chosen operator from the predefined set $\mathcal{O}$. Each operator specifies how new child nodes are created, how semantic content is assigned, and how parent–child dependencies are maintained. By repeatedly applying this procedure, the framework incrementally grows a structured tree in which intermediate values and dependencies define the agent's reasoning path. To capture different reasoning paradigms, we support two complementary modes for constructing the tree: a bottom-up mode that reflects reasoning from evidence to conclusions, and a top-down mode that mirrors task decomposition from goals to subgoals. We detail both modes below and illustrate them in Algorithm 1.

**Bottom-Up Generation Mode.** In the bottom-up mode, the dependency tree is constructed by progressively defining parent nodes from the information of their children. At each iteration, a leaf node is selected and expanded using a randomly chosen operator. The operator specifies how many children to generate, and each new child is assigned unique content to ensure semantic diversity. The

operator's rule then computes a value from these children, and this value becomes the identifier of the parent node. In this way, the parent's identity is grounded in the content of its descendants, and the tree grows upward from concrete evidence to increasingly abstract variables. As indicated by the orange arrows in Figure 2, under the bottom-up generation paradigm, the data generation proceeds by first determining the content generated at the leaf nodes and subsequently assigning the IDs of their parent nodes.

**Top-Down Generation Mode.** In the top-down mode, the dependency tree is generated by progressively decomposing a parent node into its children. At each iteration, a leaf node is selected and expanded using a randomly chosen operator. The operator assigns unique identifiers to the new child nodes and then generates a rule that references these identifiers; this rule is stored as the content of the parent node. Unlike the bottom-up process, where a parent's identity is derived from the concrete values of its children, here the parent's content encodes a dependency that will only be resolved once the children are instantiated with values. In this way, the tree grows downward from abstract goals to concrete subgoals, mirroring a deductive reasoning process. As indicated by the green arrows in Figure 2, under the top-down generation paradigm, the data-generation path proceeds by first finalizing the content of the main module and then decomposing it into subordinate modules.

**Leaf Consolidation.** To manage the growth of the dependency tree, we introduce a consolidation mechanism that controls branching width and encourages deeper chains of reasoning. Whenever the number of leaf nodes exceeds a predefined threshold, a consolidation step may be probabilistically triggered. This step reduces the number of active leaves by aggregating or redirecting them, thereby preventing uncontrolled expansion of the tree. Although the purpose is the same across both modes, the semantics differ. In the bottom-up setting, consolidation aggregates several leaves into a new node whose content is defined as a list of their identifiers, effectively packaging multiple pieces of evidence into a single representation. In the top-down setting, consolidation instead redirects several leaves to reference a newly created node with a fresh identifier, enforcing a shared dependency among them. In both cases, a potentially large set of leaves is replaced by a single node, narrowing the breadth of the tree while extending its depth, and thus creating longer dependency chains for the agent to resolve.

**Finalization and Task Definition.** The construction continues until a fixed number of operations $N$ is reached, after which the tree is finalized into a complete task. In the bottom-up mode, this involves assigning unique identifiers to all remaining leaves and instantiating the root with a random value, so that the agent must infer the root value from the set of leaf identifiers via exploring the dependency tree. In the top-down mode, each leaf is instead instantiated with a random value consistent with its type, and the values of internal nodes, including the root, are recursively computed by applying the stored rules; the agent's task is to recover the variable contained in the root. This finalization ensures that both modes yield well-posed reasoning problems with clearly defined inputs and a unique target output, providing a systematic way to evaluate long-horizon reasoning under error accumulation.

### 3.3 Constructing LORE Benchmark through TaskWeaver Instantiations

To demonstrate the flexibility of our framework and to construct a comprehensive benchmark, we instantiate the task generation process in multiple domains. Each instantiation specifies how nodes are represented, which operators are available, and what modalities are involved, while reusing the same dependency-tree construction principles described earlier. By doing so, we obtain benchmark tasks that capture complementary dimensions of long-horizon agentic reasoning: document understanding and navigation, perception-reasoning integration in multi-modal settings, and structured logic under execution constraints in code. These domain-specific instantiations provide diverse yet systematic evaluation of an agent's robustness to long-horizon reasoning. Illustrative examples from these domains are shown in Figure 2. We also provide more detailed examples with their ground truth solution trajectory in Appendix B.

**Document Understanding Navigation.** In this instantiation, we simulate a structured document system using the bottom-up generation mode. Each node corresponds to a document that stores either numeric or string values, or a rule specifying how new values are derived from existing ones. The operator set $\mathcal{O}_{\text{doc}}$ includes arithmetic operation (e.g., $o_{\text{add}}$, $o_{\text{sub}}$) and string operations (e.g., $o_{\text{concat}}$), and can be extended with more advanced functions in harder variants. A distinctive aspect of this domain is that documents may contain both relevant and irrelevant fields. To succeed, the agent must parse each document, identify which information contributes to the dependency chain, disregard

---

**Algorithm 1** Task Generation Process in TaskWeaver

---

1: **Input:** Mode $m \in \{$'bottom-up', 'top-down'$\}$, Target iterations $N$, Operator set $\mathbb{O}$, Leaf threshold $\theta_{\max}$
2: Initialize tree $\mathcal{T}$ with a single root node $v_{root}$
3: **for** $i = 1$ to $N$ **do**
4:      $v_p \leftarrow$ RandomSelect(Leaves($\mathcal{T}$))
5:      $O \leftarrow$ RandomSelect($\mathbb{O}$)
6:      $k \leftarrow$ RandomInt($O.n_{\min}, O.n_{\max}$)
7:      $V_c \leftarrow \emptyset$
8:      **for** $j = 1$ to $k$ **do**
9:          $v_{c_j} \leftarrow O.$CreateChild()
10:         $V_c \leftarrow V_c \cup \{v_{c_j}\}$
11:         **if** $m = $ 'bottom-up' **then**
12:             $v_c.c \leftarrow$ GenerateUniqueContent()
13:             $v_p.id \leftarrow O.$ExecuteRule($O.$GenerateRule($V_c$), $\{v.c$ for $v \in V_c\}$)
14:         **else if** $m = $ 'top-down' **then**
15:             $v_c.id \leftarrow$ GenerateUniqueID()
16:             $v_p.c \leftarrow O.$GenerateRule($V_c$)
17:      Add children $V_c$ to $\mathcal{T}$ under $v_p$
18:      **if** $|$Leaves($\mathcal{T}$)$| > \theta_{\max}$ **and** RandomFloat() $< p_{collect}$ **then**
19:         ConsolidateLeaves($\mathcal{T}$, mode $= m$)
20: Finalize leaf IDs and root value for $\mathcal{T}$
21: **Return** $\mathcal{T}$

---

distractors, and compute expressions that resolve the identifiers of subsequent documents. This recursive navigation continues until the target document is reached and its content is returned as the answer. In this way, the instantiation distills the essence of document understanding: filtering and integrating information across multiple records under long-horizon dependencies.

**Multi-modal Understanding.** This instantiation extends the document setting by introducing visual inputs alongside symbolic content. Some nodes that originally store numerical are replaced with images of objects, and the agent must infer the value by counting the objects in the image. Other nodes are replaced with rendered formulas, requiring the agent to read and interpret symbolic expressions from visual input. In both cases, the dependency chain now mixes perception with reasoning, rather than relying solely on textual values. As illustrated in Figure 2, harder variants further increase the complexity of the visual inputs. Images may include objects of different shapes and colors, and the target value becomes the count of those matching a specified attribute combination. Formula images may also be distorted by transformations such as scaling, rotation, or shearing, making them more difficult to recognize. To succeed, the agent must apply image processing tools to extract the correct information before integrating it into the reasoning process. This instantiation therefore evaluates an agent's ability to combine perception and symbolic reasoning under long-horizon dependencies.

**Executable Code Reasoning.** This instantiation employs the top-down generation mode, ensuring that the constructed dependency tree corresponds to an executable program. In the basic setting, operators include integer arithmetic and conditional statements, and the agent must reason about the program's output by following its control flow. To increase difficulty, we introduce an asynchronous task operator that spawns multiple concurrent threads sharing a global variable. Each thread may read and update this variable in an interleaved order, creating race conditions that make the final outcome highly non-trivial to predict. While the program remains deterministic, correctly inferring its result requires an understanding of concurrency and resource competition. This domain therefore stresses the ability of agents to perform structured program reasoning under execution constraints.

## 4 EXPERIMENTS

In this section, we evaluate the performance of state-of-the-art LLMs on the LORE benchmarks generated by TaskWeaver. Our experiments are designed to answer the following questions: (1) How does the capability of state-of-the-art LLMs decay as the number of sequential, dependent tool calls

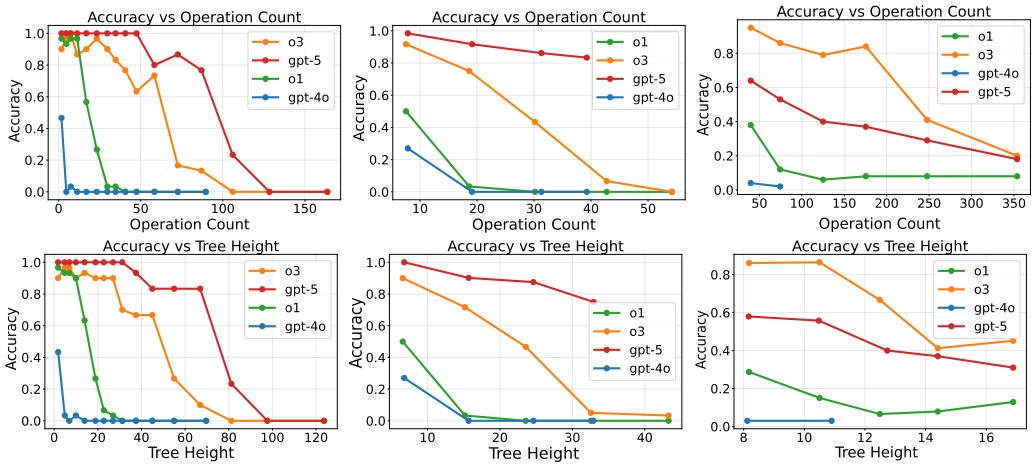

(a) Document Understanding    (b) Multi-modal Understanding    (c) Executable Code Reasoning

Figure 3: Model accuracy on LORE-Standard with respect to operation count ($N$) and tree height, evaluated across the document understanding navigation, multi-modal understanding, and executable code reasoning domains. $N$ is increased until model performance degrades to an unacceptable level.

increases? (2) How does the complexity of individual operations affect this performance decay? (3) What are the primary failure modes for LLMs on these long-horizon tasks?

## 4.1 EXPERIMENTAL SETUP

**Models.** We evaluate a suite of four powerful, proprietary models known for their strong reasoning capabilities: GPT-4o (OpenAI, 2024), GPT-5 (OpenAI, 2025a), o1 (OpenAI, 2025b), and o3 (OpenAI, 2025c). This selection allows us to analyze performance across a spectrum of state-of-the-art models and understand how advanced reasoning capabilities correlate with robustness on long-horizon tasks.

**Benchmark Datasets.** We generate all evaluation tasks using the TaskWeaver framework. First, we establish a "standard" benchmark called LORE-Standard for each of our three domains: Document Understanding Navigation, Multi-modal Understanding, and Executable Code Reasoning. To specifically investigate how increased per-operation complexity affects the rate of performance decay over long interaction sequences, we then introduce "hard" variants called LORE-Hard for the Multi-modal and Code domains by incorporating more challenging operators, as detailed in Section 3. For each benchmark configuration, we progressively increase the number of operations ($N$) to generate tasks of increasing length, continuing until the accuracy of the evaluated models degrades to an unacceptable level. This process resulted in a distribution of tasks with operation counts ranging from 1 to 350, allowing for a fine-grained analysis of performance degradation across statistically significant samples.

**Evaluation Metrics.** Our primary metric is *Accuracy (Acc.)*, defined as the percentage of tasks where the agent returns the exactly correct final answer. In addition, we analyze agent performance by plotting accuracy against two key indicators of task complexity: the *Operation Count* ($N$), which measures the total length of the reasoning chain, and the *Tree Height*, which measures the length of the longest sequential dependency chain required to reach the solution.

**Implementation Details.** To ensure a fair comparison, all models are integrated into a standardized agentic framework that follows a ReAct-style reasoning loop. The agent is provided with the necessary tools for each domain and a domain-specific prompt, which can be found in Appendix B. For all experiments, we set the model's temperature to $0.0$ to ensure deterministic outputs. To avoid artificially limiting the agent's performance on complex, long-horizon tasks, we do not set an explicit limit on the number of tokens per generation step. This ensures the model has sufficient capacity to complete its reasoning process without being cut off. In addition, the evaluation is conducted in a sandboxed environment to safely execute any generated code or tool calls. More implementation details can be found in Appendix C.

## 4.2 PERFORMANCE DECAY IN LONG-HORIZON REASONING

To evaluate how agent capabilities scale with task length, we conduct experiments on LORE-Standard. The results, shown in Figure 3, yield three main findings.

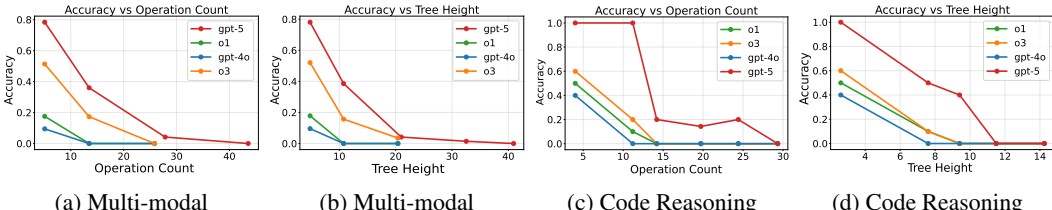

|                  |                  |                   |                   |
| :--------------: | :--------------: | :---------------: | :---------------: |
| (a) Multi-modal  | (b) Multi-modal  | (c) Code Reasoning | (d) Code Reasoning |

Figure 4: Model accuracy on LORE-Hard with respect to operation count ($N$) and tree height, evaluated in the Multi-modal Understanding and Executable Code Reasoning domains. $N$ is increased until model performance degrades to an unacceptable level.

First, a clear distinction emerges between models with strong reasoning capabilities and those without. Models such as 'GPT-5', 'o3', and 'o1' demonstrate high accuracy on short-horizon tasks. In contrast, 'GPT-4o''s performance deteriorates rapidly even on tasks with very few steps, underscoring that foundational reasoning ability is a prerequisite for tackling multi-step agentic workflows.

Second, model performance follows a consistent hierarchy. 'GPT-5' achieves the highest accuracy, followed by 'o3' and 'o1', while 'GPT-4o' lags significantly behind. This demonstrates that advances in reasoning-oriented models translate directly into more reliable performance on long-horizon tasks.

Finally, all models display a performance ceiling beyond which accuracy collapses. Even the strongest model, 'GPT-5', becomes unreliable as task length increases, with accuracy falling below 20% on tasks involving roughly 120 operations in document understanding and navigation. The effect is even more severe when measured by dependency depth: accuracy can collapse at tree heights under 80, highlighting the brittleness of current agents in sustaining long logical chains.

## 4.3 Impact of Operation Complexity on Long-Horizon Reasoning

To explore how performance decay is affected by increased per-operation complexity, we evaluate the performance on LORE-Hard, including the Multi-modal and Code domains. These benchmarks introduce more demanding operations, such as requiring agents to interpret transformed images (e.g., stretched or rotated) in the multi-modal domain, and to understand complex resource competition in asynchronous code. We re-evaluated all models on these more challenging benchmarks, with the results presented in Figure 4. The findings lead to two conclusions.

First, as per-step difficulty grows, the performance gap between moderately capable and weaker models narrows sharply. On these harder tasks, models such as 'o1' and 'o3' are quickly overwhelmed, and their performance collapses to nearly the same level as the non-reasoning baseline 'GPT-4o', which already fails on very short tasks. This indicates that effective reasoning ability is relative: once the complexity of a single operation exceeds a model's threshold, its ability to sustain multi-step planning disintegrates and its behavior converges with that of weaker systems.

Second, the operational horizon of even the strongest model, 'GPT-5', shrinks dramatically. While it remains competent on standard tasks with more than 100 operations, its accuracy on LORE-Hard falls below reliability after only about 12 operations. This steep decline highlights a critical limitation of current state-of-the-art agents: they are not yet capable of handling long-horizon reasoning when individual steps involve complex, realistic operations.

## 4.4 Error Analysis

Based on our experiments, we collected a large corpus of successful and failed agent trajectories on long-horizon tasks. A qualitative analysis of these failures reveals several recurring patterns that are not mutually exclusive but represent distinct cognitive bottlenecks in current agent architectures. We summarize these into three primary categories:

**Premature Halting due to Incomplete State Assessment.** A frequent failure mode occurs when an agent attempts to evaluate an expression or make a decision before acquiring all necessary dependencies. For instance, as illustrated in Figure 5(a), when tasked to compute $c = a + b$, an agent might have successfully retrieved the value for $a$ and identified the document containing $b$ recently, but instead of reading the document, it resorts to guessing a value for $b$ or treating it as

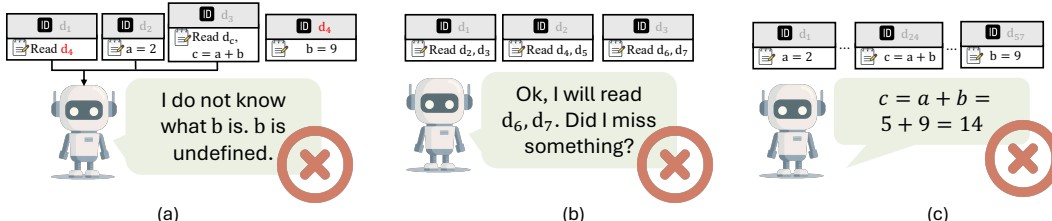

Figure 5: Illustrative examples of three primary agent failure modes. (a) Premature halting due to incomplete state assessment. (b) Information loss from ineffective working memory. (c) Degradation of reasoning over extended contexts.

undefined. After a few failed attempts, the agent often concludes that the task is unsolvable with the available information and halts, rather than re-evaluating its state and identifying the simple, unexplored step of reading the document for $b$. This behavior reflects a breakdown in the agent's ability to monitor its own knowledge state when encountering an impasse. Addressing this failure may require mechanisms that prompt the agent to consolidate what has been gathered and reassess its plan, ensuring that simple unexplored steps are not prematurely overlooked.

**Ineffective Working Memory and Information Triage.** Our benchmark tasks, much like real-world scenarios, involve highly uneven patterns of information discovery. At times, an agent may receive a large batch of new leads (e.g., file IDs) to investigate. We observed that agents typically begin exploring these leads sequentially, but, as shown in Figure 5(b), if another large batch arrives before the first is fully processed, the earlier leads are often neglected. Because the agent's attention and context window are limited, these pending leads may be displaced from memory, never resurfacing in the reasoning process. This premature loss of information is particularly damaging in long-horizon settings, where seemingly minor omissions early in the trajectory can block access to critical dependencies much later on, making the task unsolvable. These observations highlight the need for robust working memory mechanisms that can triage, buffer, and prioritize information over extended histories, ensuring that important but deferred leads are not forgotten as the interaction unfolds.

**Degradation of Reasoning over Extended Contexts.** In many failure cases, the agent successfully collects all the information and dependencies needed for the final answer. However, as shown in Figure 5(c), these values and the rules for combining them are often scattered across a long and noisy context history. When it comes time to synthesize the result, the agent fails at simple computation or logical application, incorrectly retrieving or combining values that lie far apart in its context.

This challenge arises in two ways. Sometimes it is intrinsic to the task: critical components are only discovered after many interactions, leaving them inherently distributed. Other times it is self-induced: the agent may gather all necessary values within a few turns but fail to recognize that the expression is already solvable. Instead of completing the computation, it continues exploring, pushing the relevant information deeper into its growing history. What was once a localized sub-problem thus becomes a retrieval challenge over a long and convoluted context.

This failure mode highlights the need for solutions beyond simply enlarging context windows. For example, dedicated long-term memory modules or sub-agent architectures could allow discrete computations to be offloaded to specialized agents with clean, isolated contexts.

## 5 CONCLUSION

In this paper, we introduced TaskWeaver, a controllable platform for generating benchmarks that directly assess the long-horizon reasoning abilities of LLM agents. By abstracting tool use as file-read operations, TaskWeaver isolates the core challenge of tracking state and integrating intermediate results across extended interaction sequences. Building on this framework, we instantiated tasks in document reasoning, multi-modal integration, and executable code analysis to construct LORE, a benchmark with two variants: LORE-Standard and LORE-Hard. Our evaluation shows that the state-of-the-art model performance of reasoning degrades substantially as task length and per-step complexity increase. These findings highlight long-horizon robustness as a central open challenge for advancing the next generation of agentic systems.

ETHICS STATEMENT

The authors of this paper have read and adhered to the ICLR Code of Ethics. Our work introduces a framework for procedurally generating benchmark tasks for evaluating LLM agents. All data generated and used in our experiments is synthetic and does not contain any personally identifiable information (PII) or sensitive real-world data, thus avoiding privacy concerns. The tasks are designed to be abstract reasoning challenges and do not simulate or encourage harmful behaviors. We believe that by providing a controlled environment to study and measure the failure modes of agents in long-horizon tasks, our work contributes positively to the responsible development of more robust, reliable, and safer AI systems.

REPRODUCIBILITY STATEMENT

To ensure the reproducibility of our research, we commit to making all relevant artifacts publicly available upon publication. This includes the source code for the TaskWeaver generation framework, all evaluation scripts, and the complete datasets used in our experiments. A detailed description of our methodology, including the unified generation algorithm (Algorithm 1) and domain-specific instantiations, is provided in Section 3. The experimental setup, including the models, frameworks, and primary evaluation metrics, is described in Section 4. Further details, including comprehensive task examples and specific agent prompts, will be provided in the Appendix B to allow for the full replication of our results.

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

## A  LLM USAGE FOR PAPER WRITING

We have used Large Language Models (LLMs) for the limited purpose of language polishing. The core aspects of this paper, including the initial ideation, structural framework, and primary writing, were completed entirely by human authors. All text polished by LLMs was subsequently reviewed, edited, and, when necessary, rewritten by the human authors to ensure the accuracy and integrity of the content. The human authors are fully responsible for the research design, experiments, analysis, and the final version of this work.

## B  DETAILED TASK EXAMPLES

This appendix section offers detailed, step-by-step examples of generated tasks for each of the three domains. Each example includes the full prompt, the available tools, the environment state (i.e., document contents), and a ground truth solution trajectory that illustrates the intended reasoning path.

### B.1  DOCUMENT UNDERSTANDING NAVIGATION EXAMPLE

This task, generated using the bottom-up mode, requires the agent to navigate a web of interconnected documents. The agent must parse information, perform calculations to derive the IDs of subsequent documents, and follow the dependency chain to find the final answer.

#### B.1.1  TASK SPECIFICATION

The example is illustrated in Table 1.

#### B.1.2  AVAILABLE TOOLS

`read_document(file_id: str) -> str` This tool takes a string 'file_id' as input and returns the raw text content of the corresponding document.

Table 1: A Document Understanding Navigation Example

| Component | Value |
|---|---|
| **Prompt** | You are a research assistant working with a document database. Your task is to find the value of varible 'v0' by reading and analyzing the provided documents. Start by examining documents: v10%d, v11%U, v12%HxA, v13%Zcw, v14%TqiU, v15%TeM, v16%OIFK, v17%QMXl. Use the read_document tool to access each document and follow any references or calculations to find the final answer. |
| **Operation Count** | 2 |
| **Tree Height** | 2 |
| **Ground Truth** | XUyWqrar |

### B.1.3 ENVIRONMENT: DOCUMENT CONTENTS

```
v4%186: "v0: XUyWqrar."
v9%kLVvGzgVbD: "Parameter v1 is set to 44."
v10%d: "v2: 46."
v11%U: "v3: 96."
v12%HxA: "Read the document 'v4%X' for more information, where the X is
          the value of the expression v1 + v2 + v3. Note that you
          should use the negative sign if X is negative, but do not
          use the positive sign if X is positive or zero."
v13%Zcw: "Field v5 contains vGz."
v14%TqiU: "v6 = D."
v15%TeM: "Parameter v7 is set to kLV."
v16%OIFK: "v8 has value gVb."
v17%QMXl: "Read the document 'v9%X' for more information, where the X is
          the value of the expression v7 + v5 + v8 + v6. Each variable
          in the expression should be treated as a string and the
          operator + is used to concatenate the strings."
```

### B.1.4 GROUND TRUTH SOLUTION TRAJECTORY

An ideal agent would perform the following sequence of reasoning and actions:

1. **Initial Analysis:** The prompt provides eight starting document IDs. I will read all of them to gather initial information and find rules for proceeding.

2. **Action:** Call 'read_document' on all eight initial IDs: 'v10%d', 'v11%U', 'v12%HxA', 'v13%Zcw', 'v14%TqiU', 'v15%TeM', 'v16%OIFK', 'v17%QMXl'.

3. **Observation & Thought:**
   - **Acquired Variables:**
     - From 'v10%d': 'v2 = 46'
     - From 'v11%U': 'v3 = 96'
     - From 'v13%Zcw': 'v5 = "vGz"'
     - From 'v14%TqiU': 'v6 = "D"'
     - From 'v15%TeM': 'v7 = "kLV"'
     - From 'v16%OIFK': 'v8 = "gVb"'
   - **Acquired Rules:**
     - From 'v12%HxA': (Rule 1) The ID for the next document is 'v4%X', where 'X = v1 + v2 + v3'.
     - From 'v17%QMXl': (Rule 2) The ID for the next document is 'v9%X', where 'X = v7 + v5 + v8 + v6' (string concatenation).
   - **Rule Computability Assessment:**
     - Rule 1 requires 'v1', 'v2', 'v3'. I have 'v2' and 'v3', but 'v1' is missing. This rule is **not yet computable**.

      – Rule 2 requires 'v7', 'v5', 'v8', 'v6'. I have all of these values. This rule is **computable**.

- **Rule Execution:** I will execute Rule 2.
  - 'X = "kLV" + "vGz" + "gVb" + "D" = "kLVvGzgVbD"'.
  - The new document ID is 'v9%kLVvGzgVbD'.
- **Next Step:** I will read document 'v9%kLVvGzgVbD' to find the value of 'v1'.

4. **Action:** 'read_document(doc_id='v9%kLVvGzgVbD')'.

5. **Observation & Thought:**
   - **Acquired Variables:**
     - From 'v9%kLVvGzgVbD': 'v1 = 44'.
   - **Acquired Rules:** None from this read.
   - **Rule Computability Assessment:** I will re-evaluate my known rules.
     - Rule 1 requires 'v1', 'v2', 'v3'. I now have all three values: 'v1=44', 'v2=46', 'v3=96'. This rule is **now computable**.
   - **Rule Execution:** I will execute Rule 1.
     - 'X = 44 + 46 + 96 = 186'.
     - The new document ID is 'v4%186'.
   - **Next Step:** I will read document 'v4%186'.

6. **Action:** 'read_document(doc_id='v4%186')'.

7. **Observation & Thought:**
   - **Acquired Variables:**
     - From 'v4%186': 'v0 = "XUyWqrar"'.
   - **Acquired Rules:** None.
   - **Rule Computability Assessment:** I have found the target variable 'v0'. The task is complete.

8. **Final Answer:** The value of variable 'v0' is `XUyWqrar`.

### B.2 MULTI-MODAL UNDERSTANDING EXAMPLE

This task asks an agent to traverse multimodal documents (text + images), extract variables and rules from text, derive numeric values from images, and construct new document IDs. If images are distorted, the agent uses transformation tools (`rotate_image`, `flip_image`, `stretch_image`) to restore readability, then follows the dependency chain to reach the final answer.

#### B.2.1 TASK SPECIFICATION

The example is illustrated in Table 2.

Table 2: A Multi-modal Understanding Example

| Component | Value |
|---|---|
| Prompt | You are a research assistant working with a document database. Your task is to find the value of varible 'v0' by reading and analyzing the provided documents. Start by examining documents: v8%Eiv, v9%R, v10%h, v11%VS, v12%ZId, v13%a. Use the read_document tool to access each document and follow any references or calculations to find the final answer. |
| Operation Count | 2 |
| Tree Height | 2 |
| Ground Truth | `SyVssiFG` |

#### B.2.2 AVAILABLE TOOLS

**read_document(file_id: str) -> str** The input is `file_id`, which specifies the target document. The output is the raw text content of that document as a `str`.

**rotate_image_tool(doc_id: str, angle: float, documents: dict)**
**-> str** The inputs are `doc_id` (a multimodal document ID), `angle` (in degrees), and `documents` (the document store). The output is a description string containing the rotated image encoded in base64.

**flip_image_tool(doc_id: str, flip_type: str, documents: dict)**
**-> str** The inputs are `doc_id` (a multimodal document ID), `flip_type` {`horizontal`, `vertical`}, and `documents` (the document store). The output is a description string containing the flipped image encoded in base64.

**stretch_image_tool(doc_id: str, width_factor: float,**
**height_factor: float, documents: dict) -> str** The inputs are `doc_id` (a multimodal document ID), `width_factor` and `height_factor` (positive floats), and `documents` (the document store). The output is a description string containing the scaled image encoded in base64.

### B.2.3 ENVIRONMENT: MULTIMODAL DOCUMENT CONTENTS

```
v8%Eiv: "v1 has value VcL."
v9%R: "v2: mQs."
v10%h: {
  "text": "The value of v4 is the number of cyan circles in the image.",
  "image_path": "complex_geo_0001.png"
}
v11%VS: {
  "text": "The value of v5 is the number of brown squares in the image.",
  "image_path": "complex_geo_0002.png"
}
v12%ZId: {
  "text": "The value of v6 is the number of blue squares in the image.",
  "image_path": "complex_geo_0003.png"
}
v13%a: {
  "text": "1. IDENTIFY the mathematical expression shown in the image
           2. CALCULATE the numerical value of that expression
           3. CONSTRUCT the document ID as 'v7%X' where X
           is the calculated value
           4. READ the 'v7%X' using the constructed ID
           IMPORTANT: Use negative sign (-) for negative values,
           no sign for positive/zero values.
           NOTE: The current image may have been transformed.
           If you find the formula in the image is unreadable
           or distorted, you can directly call transformation tools
           (rotate_image, flip_image, stretch_image) in the
           next step to restore the image,
           instead of using the read_document tool.
           If the formula is clear and readable, you can directly use it.",
  "image_path": "transformed_expr_0001.png"
}
v7%-7: "Read the document 'v3%X' for more information,
where the X is the value of the expression v2 + v1.
Each variable in the expression should be treated as
a string and the operator + is used to concatenate the strings."
v3%mQsVcL: "Value of v0 is SyVssiFG."
```

### B.2.4 GROUND TRUTH SOLUTION TRAJECTORY

An ideal agent would perform the following sequence of reasoning and actions:

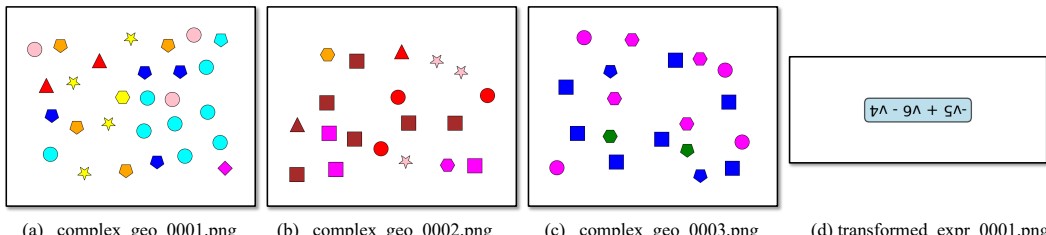

(a) complex_geo_0001.png    (b) complex_geo_0002.png    (c) complex_geo_0003.png    (d) transformed_expr_0001.png

Figure 6: The pictures of the multimodal document example.

1. **Initial Analysis:** The prompt instructs me to find `v0` by reading multimodal documents and following references. The starting IDs in this environment are: `v8%Eiv`, `v9%R`, `v10%h`, `v11%VS`, `v12%ZId`, `v13%a`. I will read all six to collect variables and navigation rules.

2. **Action:** Call `read_document` on `v8%Eiv`, `v9%R`, `v10%h`, `v11%VS`, `v12%ZId`, `v13%a`.

3. **Observation & Thought:**
   - **Acquired Variables (text nodes):**
     - From `v8%Eiv`: `v1 = "VcL"`.
     - From `v9%R`: `v2 = "mQs"`.
   - **Image-derived Variables:**
     - From `v10%h`: The value of `v4` equals the number of cyan circles in the image `complex_geo_0001.png`. I will inspect the image to count cyan circles.
     - From `v11%VS`: The value of `v5` equals the number of brown squares in `complex_geo_0002.png`.
     - From `v12%ZId`: The value of `v6` equals the number of blue squares in `complex_geo_0003.png`.
   - **Rule from the Expression Image:**
     - From `v13%a`: Identify the mathematical expression shown in `transformed_expr_0001.png`, compute its numeric value $X$, construct document ID `v7%X` (negative with '-', non-negative without '+'), and read it. The note indicates the image may be transformed, so I must verify readability and, if needed, restore it using `rotate_image`, `flip_image`, or `stretch_image`.

4. **Action (Image Reading for Counts):**
   - Inspect `complex_geo_0001.png` for cyan circles and set `v4` to the observed count.
   - Inspect `complex_geo_0002.png` for brown squares and set `v5` to the observed count.
   - Inspect `complex_geo_0003.png` for blue squares and set `v6` to the observed count.

5. **Observation & Thought (Counts Outcome):**
   - The dataset's intended counts yield: `v4 = 8`, `v5 = 6`, `v6 = 7`.

6. **Action (Expression Image Readability Check):** Inspect `transformed_expr_0001.png`. The filename suggests a 180-degree rotation; if the formula appears upside down or unreadable:
   - Call `rotate_image_tool(doc_id='v13%a', angle=180, documents)` to restore the upright orientation.
   - If further correction is needed, optionally apply `flip_image_tool` or `stretch_image_tool` minimally to improve readability.

7. **Observation & Thought (Expression Identification):**
   - After restoring orientation, identify the numeric expression in the image and compute $X$. The environment indicates that the computed value corresponds to the ID `v7%-7`, so $X = -7$.
   - Thus, I should read `v7%-7`.

8. **Action:** `read_document(doc_id='v7%-7')`.

9. **Observation & Thought (String Concatenation Rule):**

- From `v7%-7`: Read document `'v3%X'` where $X = v2 + v1$, treating variables as strings and using the '+' operator for concatenation.
- I have `v2 = "mQs"` and `v1 = "VcL"`; therefore $X = "mQs" + "VcL" = $ `"mQsVcL"`.
- The next document ID is `v3%mQsVcL`.

10. **Action:** `read_document(doc_id='v3%mQsVcL')`.

11. **Observation & Thought (Final Variable):**

- From `v3%mQsVcL`: `v0 = "SyVssiFG"`.
- The target variable has been found; the task is complete.

12. **Final Answer:** The value of variable `v0` is `SyVssiFG`.

### B.3    EXECUTABLE CODE REASONING EXAMPLE

This task, generated using the top-down mode, requires the agent to analyze a small codebase distributed across multiple files. The agent must trace function calls and dependencies between files to determine the final return value of the main program.

#### B.3.1    TASK SPECIFICATION

The example is illustrated in Table 3.

Table 3: An Executable Code Reasoning Example

| Component | Value |
|---|---|
| **Prompt** | You are a code analyst assistant working with a codebase. Your task is to determine the return value of the main function in the main.py. Use the read_document tool to read the content of each python file and follow any references or calculations to find the final answer. For the async functions, you should treat all operations as no-time-consuming except the sleep operation. |
| **Operation Count** | 2 |
| **Tree Height** | 2 |
| **Ground Truth** | 115 |

#### B.3.2    AVAILABLE TOOLS

**`read_document(file_id: str) -> str`** This tool takes a filename (e.g., 'main.py') as input and returns the source code contained within that file.

#### B.3.3    ENVIRONMENT: CODEBASE CONTENTS

```
# main.py
import v0
import v1

def main():
    return v0.main() + v1.main()

if __name__ == '__main__':
    print(main(), end='')

# v0.py
def main():
    return 44

# v1.py
import v3
import v4
```

```
import v2

def main():
    return v3.main() + v4.main() - v2.main()

# v2.py
def main():
    return 24

# v3.py
def main():
    return 45

# v4.py
def main():
    return 50
```

### B.3.4 GROUND TRUTH SOLUTION TRAJECTORY

An ideal agent would perform the following sequence of reasoning and actions:

1. **Initial Analysis:** The goal is to find the return value of 'main()' in 'main.py'. I will start by reading 'main.py'.

2. **Action:** 'read_document('main.py')'.

3. **Observation & Thought:** The code in 'main.py' shows that its 'main' function returns the sum of 'v0.main()' and 'v1.main()'. To solve this, I need to find the return values from 'v0.py' and 'v1.py'.

4. **Action:** 'read_document('v0.py')'.

5. **Observation & Thought:** The file 'v0.py' defines a 'main' function that returns the integer '44'. So, 'v0.main() = 44'. Now I need to determine the value of 'v1.main()'.

6. **Action:** 'read_document('v1.py')'.

7. **Observation & Thought:** The file 'v1.py' shows that its 'main' function returns 'v3.main() + v4.main() - v2.main()'. This introduces three new dependencies: 'v2.py', 'v3.py', and 'v4.py'. I need to read these files to find their return values.

8. **Action:**
   - 'read_document('v2.py')'
   - 'read_document('v3.py')'
   - 'read_document('v4.py')'

9. **Observation & Thought:** From the files, I have found:
   - 'v2.main()' returns '24'.
   - 'v3.main()' returns '45'.
   - 'v4.main()' returns '50'.
   Now I can calculate the return value of 'v1.main()': $45 + 50 - 24 = 71$'.

10. **Thought:** I have both required values: 'v0.main() = 44' and 'v1.main() = 71'. I can now calculate the final result from 'main.py'.

11. **Calculation:** The final result is $44 + 71 = 115$.

12. **Final Answer:** The return value of the main function is 115.

## C IMPLEMENTATION DETAILS

**API and Model Configuration.** All experiments were conducted using the Azure OpenAI service. We accessed the four evaluated models, GPT-4o, GPT-5, o1, and o3, via the 2025-01-01-preview API version. For the models that support it (GPT-4o and GPT-5), we

enabled the parallel tool calling feature to allow the agent to execute multiple tool calls concurrently within a single turn.

**Failure Handling and Loop Prevention.** To prevent agents from becoming stuck in non-productive loops, we implemented a failure-handling mechanism based on consecutive tool call failures. We define a "failed tool call round" as a turn in which the agent attempts one or more tool calls, and every single one of them fails (e.g., due to incorrect parameters or hallucinated tool names). We maintain a counter for these consecutive failures. If a turn includes at least one successful tool call, this counter is reset to zero. However, if an entire round of tool calls fails, the counter is incremented by one. In our experiments, we allowed a maximum of two consecutive failed rounds. If an agent failed for a third consecutive round, the task was terminated and marked as a failure. This mechanism ensures that agents are given a fair chance to recover from transient errors while preventing infinite loops that would consume excessive resources.

