# OpenReview forum: "The Agent's Marathon: Probing the Limits of Endurance in Long-Horizon Tasks"
_ICLR.cc/2026/Conference — Submitted to ICLR 2026_

### Official Review · Reviewer_bv3E · 2025-10-24

**Soundness:** 2
**Presentation:** 3
**Contribution:** 3
**Rating:** 4
**Confidence:** 4

**Summary:**

This paper introduces TaskWeaver, a rule-based and controllable task generation framework designed to systematically evaluate the long-horizon reasoning capabilities of large language model (LLM) agents.
Building upon this framework, the authors construct the LORE benchmark, which encompasses three subdomains: document understanding and navigation, multimodal information integration, and executable code analysis.
TaskWeaver enables precise control over task length and difficulty through a formal dependency tree and a configurable operator set, allowing systematic investigation of model robustness in long sequential reasoning.
Experimental results show that the performance of mainstream models, including the GPT series, declines sharply as task steps and complexity increase—accuracy approaches zero for tasks exceeding roughly one hundred steps.
Overall, the work provides a promising framework and methodology for controllable evaluation of long-horizon reasoning in intelligent agents.

**Strengths:**

- The paper is generally well structured, with a clear methodological exposition that adequately describes the design principles of the benchmark generation framework and the experimental process.
- The work targets a core problem in current agent research—evaluating long-horizon reasoning under controllable and scalable settings—which aligns well with recent research trends in agent benchmarking.
- The proposed LORE benchmark covers multiple task types, including document understanding and navigation, multimodal information integration, and executable code analysis, demonstrating good diversity and representativeness in task design.
- The framework exhibits strong theoretical generality, supporting various task structures and difficulty configurations, thereby laying a solid foundation for more systematic studies of long-horizon reasoning robustness in intelligent agents.

**Weaknesses:**

- Questionable validity of the core assumption. The paper abstracts all tool calls as “file-read” operations. While this design makes task generation more controllable and formally analyzable, it also greatly simplifies the real-world complexity of API interactions, state changes, and error propagation. This abstraction departs from realistic multi-tool or industrial settings, potentially weakening the external validity of the evaluation. The authors should more clearly articulate the benefits and trade-offs of this abstraction, quantify the gap between this simplification and real-world scenarios, and explain why this gap does not materially affect their main conclusions.
- Lack of systematic validation on framework stability and data reliability. The paper mainly presents successful examples of TaskWeaver’s task generation but does not discuss cases of potential failure or robustness limits. For instance, can the generation process produce unsolvable or non-unique tasks? How is the correctness and verifiability of generated data ensured? If abnormal or erroneous samples occur, is there an automated detection or filtering mechanism? These factors are critical to the benchmark’s trustworthiness and reproducibility.
- Insufficient methodological description, affecting reproducibility.
  - The predefined operator set is only briefly mentioned in individual domains and lacks a unified, fine-grained definition or illustrative examples. The paper does not explain how different operators control task complexity or compose hierarchical task structures.
  - High reproducibility risk. Although some examples are provided in the appendix, they are not sufficient to fully reproduce the results. The framework’s parameters, task instances, and generation strategies are not systematically documented.
  - Limited model coverage. The evaluation includes only four proprietary GPT models, with no open-source or reproducible baselines for comparison, making it difficult to assess the generality of the framework across architectures and scales.
- Experimental analysis leaves multiple open questions.
  - The cause of performance collapse with increasing task length is not fully analyzed—whether it stems from context-length overflow, memory degradation, or nonlinear reasoning-path expansion.
  - The definition of “per-step complexity” is unclear. It seems to depend on operation count and difficulty but lacks a formal metric or clear growth pattern (linear vs. nonlinear).
  - The attribution of performance degradation remains ambiguous. It is unclear whether declines arise from weakened long-term memory, limited reasoning ability, or the intrinsic design of the task structure.
  - The representativeness and sufficiency of the three chosen subdomains need justification. Why only document, multimodal, and code? Do these domains adequately cover the key dimensions of long-horizon reasoning? Are they too few or redundant? A clear rationale or ablation would help.
- Insufficient depth in related work comparison. Although the paper cites recent benchmarks such as AgentBoard and HiAgent, it lacks systematic comparison. The authors should explicitly discuss how their approach differs in evaluation objectives, task structure, and memory management mechanisms, and clarify the respective strengths and weaknesses to strengthen the positioning and novelty of the paper.

**Questions:**

See the "Weaknesses" section.

---

### Official Review · Reviewer_ycMr · 2025-10-28

**Soundness:** 3
**Presentation:** 2
**Contribution:** 3
**Rating:** 4
**Confidence:** 4

**Summary:**

This paper presents TaskWeaver, a controllable, rule-based framework for generating long-horizon reasoning benchmarks for LLM agents, addressing the lack of systematic evaluation of reasoning endurance and error accumulation. Building on this framework, the authors construct LORE (Long-horizon Reasoning Evaluation), which spans three domains. TaskWeaver abstracts tool use as file-read operations, enabling precise control over task length and complexity without confounding API differences. Extensive experiments on state-of-the-art models show that accuracy sharply deteriorates as task length or per-step complexity increases. The analysis reveals critical failure modes such as premature halting, ineffective working memory, and context degradation, underscoring that long-horizon robustness remains a central open challenge for the next generation of agentic systems.

**Strengths:**

1. The paper proposes TaskWeaver, a systematic and controllable framework for generating long-horizon reasoning tasks, effectively addressing the lack of standardized benchmarks for evaluating agent performance in extended reasoning scenarios.
2. The framework is highly abstract and generalizable, employing a file-read abstraction to unify diverse task types and eliminate confounding factors introduced by API dependencies or tool heterogeneity. Building upon this design, the LORE benchmark encompasses three complementary domains including document understanding, multi-modal integration, and executable code reasoning, providing a comprehensive and well-structured evaluation suite.
3. The experimental setup is rigorous, systematically assessing the reasoning stability and performance degradation patterns of several state-of-the-art models such as GPT-4o, GPT-5, o1, and o3. Moreover, the paper provides a detailed error analysis that identifies three major failure modes, namely premature halting, ineffective working memory, and context degradation, offering valuable insights for improving long-term memory and control in future models.
4. The paper is clearly written, logically structured, and well-organized, with the experimental and analytical sections presented in a coherent and layered manner, demonstrating strong overall quality.

**Weaknesses:**

1. The paper mainly focuses on the task generation and evaluation framework itself, lacking substantive exploration of model improvements or strategies to mitigate long-horizon degradation.
2. Although TaskWeaver provides controllable task construction, there remains a gap between the generated tasks and the semantic complexity of real-world scenarios, limiting the realism and representativeness of the benchmark.
3. While the experimental results are systematic, the analysis remains largely descriptive, with insufficient quantitative or visual comparisons across different failure types.
4. The paper does not provide an open-source implementation or sample dataset for the task generation framework, which weakens reproducibility and hinders community validation.
5. All compared models are closed-source APIs, and the absence of evaluations on open-source models such as Llama or Qwen reduces the comprehensiveness and generality of the experimental findings.

**Questions:**

1. Have the authors analyzed the agent’s intermediate strategies during reasoning (e.g., ReAct), and if so, what insights emerge about their effectiveness over extended horizons?
2. While the file-read abstraction removes API heterogeneity, does it also weaken the correspondence between the benchmark tasks and real-world applications?
3. Given the pronounced performance decay with increasing task length, did the authors investigate collapse thresholds or phase-transition–like characteristics in the failure dynamics?
4. For the three primary error types (premature halting, working-memory deficiencies, and context degradation), are there quantitative prevalence statistics or visual summaries?
5. Since LORE is evaluated only on closed-source GPT-series models, can open-source models (e.g., Llama, Qwen) be included to improve comparability and generality?
6. Do TaskWeaver’s generation rules risk over-structuring tasks in ways that limit generalization to open-ended, real-world scenarios?
7. Where are the implementation details for the asynchronous task operator in the Executable Code Reasoning setting formally specified?
8. In Figure 3, why does o3 outperform GPT-5 on the Executable Code Reasoning tasks, and what factors might explain this inversion relative to other domains?

---

### Official Review · Reviewer_tjCe · 2025-10-28

**Soundness:** 1
**Presentation:** 2
**Contribution:** 2
**Rating:** 2
**Confidence:** 4

**Summary:**

This paper introduces Task Weaver, a framework for procedurally generating benchmark tasks to test the long-horizon reasoning abilities of LLM agents. It addresses the problem that agent performance rapidly degrades over long interaction sequences, a gap not well-covered by existing benchmarks. The framework's core idea is to abstract all tool use as simple "file-read" operations , which isolates an agent's core reasoning and state-tracking abilities from superficial API complexities. The authors use this framework to create the LORE benchmark, which is instantiated across document, multi-modal, and code reasoning domains. Empirical results demonstrate that even the strongest models' performance collapses as task length and complexity increase, with accuracy approaching zero on tasks exceeding 120 steps, highlighting long-horizon robustness as a central open challenge.

**Strengths:**

1. The paper introduces a procedural and controllable benchmark generator, Task Weaver , which allows for the precise generation of tasks with adjustable horizon length and logical depth.
2. The paper abstracts all tool use as a simple "file-read" operation. This design removes superficial API complexities and allows the benchmark to directly test an agent's ability over long sequences.

**Weaknesses:**

1.This paper's core abstraction creates an artificial benchmark that lacks real-world fidelity. The framework intentionally removes "superficial API complexities" and avoids challenges like memorizing API syntax or handling heterogeneous interfaces. While this isolates the variable of reasoning length, it also means the benchmark fails to evaluate an agent's robustness to the messy, unpredictable, and error-prone nature of real-world tools and environments, which is a key part of the "long-horizon" challenge.

2. The evaluation is exclusively limited to proprietary models, neglecting the open-source landscape. The experiments evaluate only
four powerful, proprietary models (GPT-4o, GPT-5, o1, and o3). This is a significant omission, as it provides no insight into how leading open-source models (like the Llama or Qwen families) or other closed-source models (like Claude or Gemini) perform on these long-horizon tasks. This limits the benchmark's utility and relevance for the broader research community that relies on and builds upon these models.

3. The benchmark conflates task length with task complexity, as the individual steps remain simplistic. The "file-read" abstraction boils tasks down to solving dependency graphs. Even in the instantiated domains, the core operations are basic: arithmetic and string concatenation in "Document Understanding" , or simple call-tracing in "Code Reasoning". The "LORE-Hard" variants only add minor complexity (e.g., image distortion or race conditions ) rather than fundamentally complex reasoning. The benchmark primarily tests endurance at simple tasks, not the ability to perform complex reasoning over a long horizon.

4. The "file-read" abstraction oversimplifies the task, turning it into a symbolic reasoning puzzle rather than a true test of agentic planning. By abstracting all tool use into a uniform "file-read" operation, the benchmark removes core agentic challenges like tool selection, handling API errors, or planning in an environment with side effects. The agent's task is not to decide what tool is appropriate, but simply to execute a predefined (though long) dependency tree. This makes it a strong test of working memory and state-tracking, but a weak test of the autonomous planning and decision-making capabilities implied by the term "LLM agent."

5. The benchmark's instantiated domains suffer from limited representativeness. While the paper instantiates three domains, their design remains limited in applicability . For example, the "Executable Code Reasoning" task appears to be primarily about tracing function call graphs and executing simple arithmetic operations. This feels more like a symbolic manipulation task rather than a genuine test of code analysis, which in the real world involves understanding complex logic, side effects, and the intent of the code.

6. The framework's design places an overly strict reliance on exact-match symbolic File IDs. Based on the task examples, the agent's success hinges on its ability to generate and use precise file IDs. For instance, in the Document Understanding example, the agent must derive the exact ID v9%kLVvGzgVbD via string concatenation to use the read_document tool, and must perfectly calculate 186 to construct and read v4%186 . This reliance on exact symbolic matching makes the tasks more akin to "symbolic computation" rather than "robust tool use," which in the real world often involves handling fuzzy matching, typos, or ambiguity.

**Questions:**

1.The paper's experimental validation is limited to a narrow, proprietary set of models. The evaluation only includes "four powerful, proprietary models", namely GPT-4o, GPT-5, o1, and o3. This is a significant omission, as it provides no insight into how leading open-source or other closed-source model families perform on these long-horizon tasks, limiting the benchmark's utility and relevance for the broader research community.

2.The "Multi-modal Understanding" domain seems to be a superficial addition rather than a test of deep integrated reasoning. The task still relies on the simple arithmetic/string-based dependency graph from the document domain. The only change is that some leaf nodes are replaced with images where the value must be counted (e.g., number of cyan circles) or read (e.g., a rendered formula). This doesn't test complex vision-language integration; it's a simple (Perception -> Value) lookup, which is then fed into the same text-based procedural task.

3.The benchmark's core setup may be unfairly penalizing models for retrieval failures rather than reasoning failures. The "file-read" abstraction forces all intermediate results into a single, growing context history. The key failure mode, "Degradation of Reasoning over Extended Contexts" , describes an agent failing at simple arithmetic (c = a + b) when the values are scattered across a long and noisy context history. This seems less like a reasoning collapse and more like a retrieval failure that is an artificial product of the benchmark's design, which a real agent would mitigate using external memory or a structured scratchpad.

---

### Official Review · Reviewer_riHf · 2025-11-01

**Soundness:** 2
**Presentation:** 2
**Contribution:** 3
**Rating:** 2
**Confidence:** 3

**Summary:**

The paper introduces TaskWeaver, a controllable generation framework that turns every tool call into a standardized file read and uses formal operators to grow dependency trees, letting the authors synthesize arbitrarily long, verifiable tasks. Using this, the paper builds the LORE dataset. It covers three domains: document navigation, multimodal reasoning, and executable code analysis. The paper go on to conduct experiments that show steep performance decay with length and depth on various LLMs.

**Strengths:**

Treating tool use as file reads is a novel approach that strips away api idiosyncrasies and directly targets state tracking over long dependency chains.

The coverage of text navigation, multimodal perception-reasoning, and code execution is comprehensive testing of agent’s long horizon abilities in various domains of importance.

The construction of benchmark using dependency trees expanded by formal operators produce controllable, verifiable tasks with adjustable arity and depth.

The variation of difficulty level using both total operation count and per step complexity is inspiring.

**Weaknesses:**

All tested models come from the GPT family. It would be great to see the performance from different family of models, such as Claude, Gemini, and open-weight models such as DeepSeek, and models of different sizes.

The logic is clear on how authors concatenate tasks with both bottom up and top down approach with the help of a set of task expansion rules. However, it isn’t immediately clear how the task expansion rules are created, including their meaning.

In Fig.4, it appears that gpt-4o and o1 performs below 20% in multimodal setting when in low operation count. The gap between o1/gpt-4o and the rest of models is much larger on multimodal setting compared to the gap in code reasoning setting. This difference is not fully captured by the author’s claim of the difference between strong/weak reasoning capabilities of models.

In addition, at an operation count of close to 1, tasks are close to single step and it isn’t clear if the impact of “horizon” still plays a role. It would be helpful to better isolate out the factor of operation count from others. For instance, how do models perform when operation count is exactly one?

Has the author considered the impact of long context and memory on models’ performance on these tasks. The paper mentions the possibility of information being displaced due to limited context window. It would be helpful to provide more analysis such as how many memory related errors happen because information was displaced due to exceeding context window.

**Questions:**

Listed above.

---

### Meta-Review · Area_Chair_7EyL · 2026-01-05

**Summary:**

The paper introduces TaskWeaver, a framework for procedurally generating benchmark tasks to test the long-horizon reasoning abilities of LLM agents.

All reviewers gave negative comments, and since no rebuttals were provided by the authors, the AC would recommend rejection.

**Reviewer Concerns:**

* Reviewer riHf noted the limited diversity of tested models, which were restricted to the GPT family, and requested more clarity on the creation and meaning of task expansion rules. He/she also suggested better isolating the "horizon" factor from operation count and providing more detailed analysis of errors caused by context window limitations.
* Reviewer tjCe argued that the "file-read" abstraction creates an artificial benchmark lacking real-world fidelity by removing challenges like tool selection and API error handling. He/she further criticized the exclusive evaluation of proprietary models and noted that tasks rely too heavily on exact-match symbolic File IDs.
* Reviewer ycMr highlighted a lack of strategies for mitigating long-horizon degradation and noted a significant gap between the synthetic tasks and real-world semantic complexity. Additionally, he/she cited the absence of an open-source implementation and the exclusion of open-source model families as major weaknesses.
* Reviewer bv3E questioned the external validity of the core "file-read" abstraction and cited a lack of systematic validation regarding data reliability and task solvability. He/she also identified high reproducibility risks due to insufficient methodological descriptions and limited model coverage beyond the GPT series.

**Reviewer Scores:**

No rebuttals were provided, and no reviewer should change their scores.

---

### Decision · Program_Chairs · 2026-01-26

Reject